

# Resveratrol ameliorates 2,4-dinitrofluorobenzene-induced atopic dermatitis-like lesions through effects on the epithelium

Sule Caglayan Sozmen[1], Meral Karaman[2], Serap Cilaker Micili[3], Sakine Isik[1], Zeynep Arikan Ayyildiz[1], Alper Bagriyanik[3], Nevin Uzuner[1] and Ozkan Karaman[1]

[1] Faculty of Medicine, Department of Pediatric Allergy and Immunology, Dokuz Eylül University, Izmir, Turkey
[2] Faculty of Medicine, Department of Microbiology, Dokuz Eylül University, Izmir, Turkey
[3] Faculty of Medicine, Department of Histology and Embryology, Dokuz Eylül University, Izmir, Turkey

Corresponding author
Sule Caglayan Sozmen,
sulecaglayan07@yahoo.com

## ABSTRACT

**Background.** Resveratrol is a natural polyphenol that exhibits anti-inflammatory effects. The aim of this study was to investigate the effects of resveratrol treatment on epithelium-derived cytokines and epithelial apoptosis in a murine model of atopic dermatitis-like lesions.

**Material and Methods.** Atopic dermatitis-like lesions were induced in BALB/c mice by repeated application of 2,4-dinitrofluorobenzene to shaved dorsal skin. Twenty-one BALB/c mice were divided into three groups: group I (control), group II (vehicle control), and group III (resveratrol). Systemic resveratrol (30 mg/kg/day) was administered repeatedly during the 6th week of the experiment. After the mice had been sacrificed, skin tissues were examined histologically for epithelial thickness. Epithelial apoptosis (caspase-3) and epithelium-derived cytokines [interleukin (IL)-25, IL-33, and thymic stromal lymphopoietin (TSLP)] were evaluated immunohistochemically.

**Results.** Epithelial thickness and the numbers of IL-25, IL-33, TSLP and caspase-3-positive cells were significantly higher in group II compared to group I mice. There was significant improvement in epithelial thickness in group III compared with group II mice ($p < 0.05$). The numbers of IL-25, IL-33, and TSLP-positive cells in the epithelium were lower in group III than in group II mice ($p < 0.05$). The number of caspase-3-positive cells, as an indicator of apoptosis, in the epithelium was significantly lower in group III than in group II mice ($p < 0.05$).

**Conclusion.** Treatment with resveratrol was effective at ameliorating histological changes and inflammation by acting on epithelium-derived cytokines and epithelial apoptosis.

## INTRODUCTION

Atopic dermatitis (AD) is a chronic inflammatory skin disease primarily affecting children. Its prevalence has been steadily increasing during the last decade in both developed and

developing countries (*Bieber, 2008*). Although the etiology of AD remains obscure, interplay among immunological, environmental, and genetic factors leads to its development (*Boguniewicz & Leung, 2011*). However, no current treatment for AD can ameliorate its pathogenesis permanently.

The pathogenesis of AD has not been clearly identified, but most information about its immunological features has been obtained in recent years. It was demonstrated that the epidermal cells in AD are unique in terms of both their barrier and immunological properties. The epidermis of patients with AD exhibits significant barrier disruption and prominent keratinocyte pathology. Keratinocytes are specialized epithelial cells in skin tissue that contribute to the initiation and maintenance of the inflammatory process in AD and are capable of producing, as well as responding to, various inflammatory mediators (*Esche, De Benedetto & Beck, 2004*). Interleukin (IL)-25, IL-33, and thymic stromal lymphopoietin (TSLP) are mainly released from keratinocytes and promote the Th2-type immune response (*Brandt & Sivaprasad, 2011*). Dysregulated apoptosis of keratinocytes plays a major role in the pathogenesis of AD by causing spongiosis and intercellular edema, leading to impaired epithelial integrity (*Trautmann et al., 2000*).

Resveratrol is a naturally occurring polyphenol found in various types of fruits and vegetables, most notably in the skin of red grapes. Several studies indicated that it exerts various pharmacological effects, such as anticancer, antioxidant, antiangiogenic, and anti-inflammatory properties (*Harikumar & Aggarwal, 2008*). In this context, we investigated the effects of resveratrol treatment on keratinocyte-derived cytokines and keratinocyte apoptosis using a murine model of 2,4-dinitrophenylbenzene (DNFB)-induced AD-like lesions.

## MATERIALS AND METHODS

### Animals

Twenty-one 6–8-week-old male BALB/c mice weighing 18–20 g, purchased from the Department of Multidisciplinary Animal Laboratory, Dokuz Eylul University (Izmir, Turkey), were used in this study. The animals were kept in hygienic macrolane cages in air-conditioned rooms under 12-h light/dark cycles for the experiment. Food and water were provided *ad libitum* in a pathogen-free laboratory in the same department. All experimental procedures complied with the requirements of the Dokuz Eylul University Animal Care and Ethics Committee (Registration number:92/2013)

### Induction of dermatitis

The induction of AD by using DNFB was established based on previous research (*Li et al., 2013*). DNFB was purchased from Sigma Chemical (St. Louis, MO, USA) and dissolved in a mixture of acetone and olive oil (4:1). AD-like skin lesions were evoked by repeated application of 100 μL of 0.5% DNFB to the shaved backs of mice during the first week for sensitization (Fig. 1). After the first week, 100 μL of 0.2% DNFB was applied twice a week for a further 4 weeks. The lesions developed at the end of the 5th week. During the 6th week, DNFB was applied once to maintain inflammation.

**Figure 1 Schematic presentation of experimental procedure.** (A)Experimental procedure in control group. (B) Experimental procedure in vehicle control group. (C) Experimental procedure in resveratrol treatment group. DNFB, 2,4-Dinitrophenylbenzene; DMSO, dimethyl sulfoxide; IL, interleukin; TSLP, thymic stromal lymphopoietin.

## Experimental schedule

The 21 BALB/c mice were randomly divided into three groups ($n = 7$ per group), as follows: group I (control), group II (vehicle control), and group III (treatment with resveratrol) (Fig. 1).

The acetone and olive oil mixture was applied to shaved back of group I (control) without DNFB in the same manner. Atopic dermatitis –like lesions were induced in Group II (vehicle control) and group III (treatment with resveratrol).

Resveratrol was given to group III (treatment with resveratrol) at a dose of 30 mg/kg/day for 7 days during the 6th week. Resveratrol was administered to each mice after dissolved

in 100 μL dimethyl sulfoxide (DMSO) in group III (*Lee et al., 2009*; *Sharma, Huq & Singh, 2014*; *Johnson et al., 2011*). Resveratrol was purchased from Sigma-Chemical (St.Louis, MO, USA). Group II (vehicle control group) was treated with 100 μl DMSO during the 6th week of experimental procedure Saline (0.9% NaCl) was administered to group I (control group) at dose of 100 μl during the 6th week. All drugs were administered via the orogastric route. The mice were weighed beginning of the experiment, at the end of the 5th and 6th week.

Animals were sacrificed by an overdose of ketamine 24 h after the last drug administration, and dorsal skin samples were obtained for histomorphological analysis.

## Evaluation of dermatitis

Severity of dermatitis was estimated macroscopically at the end of 5th and 6th weeks. The following scoring procedure was applied:0, no symptoms; 1, mild symptoms; 2, moderate symptoms; 3, severe symptoms. The dermatitis score was described as the sum of the scores for erythema/hemorrhage, edema, excoriation/erosion and scaling/dryness (*Hanifin et al., 2001*).

## Histomorphological analysis

Skin samples were placed in buffered formalin for light microscopic evaluation. After fixation, skin samples were embedded in paraffin for light microscopic evaluation and 5-μm serial sections were obtained with a rotary microtome (Leica RM2125; Leica Biosystems, Wetzlar, Germany). The samples were then stained with hematoxylin and eosin. Using these samples, general tissue features were examined and the thickness of the epithelium was measured. Photomicrographs were taken with an Olympus DP70 camera (Olympus, Tokyo, Japan), which was adapted on an Olympus BX51 model microscope (Olympus Optical, Tokyo, Japan). The photomicrographs were taken randomly from five fields of each section. A counting frame was randomly placed four times on the image analyzer system monitor, epithelial thickness was measured (UTHSCA Image Tool for Windows, version 3.0), and the average was taken.

## Immunohistochemical detection

All sections were incubated in a solution of 3% $H_2O_2$ for 5 min to inhibit endogenous peroxidase activity and then in normal serum blocking solution. Sections were incubated in a humid chamber for 18 h at 4 °C with IL-33 monoclonal antibody at 1:100 (anti-IL-33 mouse monoclonal antibody, NBP1-75516; Novus Biologicals, Littleton, CO, USA), IL-25 monoclonal antibody at 1:100 (anti-IL-25 mouse monoclonal antibody, NBP1-72027; Novus Biologicals), TSLP monoclonal antibody at 1:100 (anti-TSLP mouse monoclonal antibody, NBP1-76754; Novus Biologicals), and anti-caspase-3 antibody at 1:100 (AB3623; Millipore, Billerica, MA, USA). Sections were incubated with biotinylated IgG, followed by streptavidin conjugated to horseradish peroxidase for 15 min, each prepared in accordance with the manufacturer's instructions (85-9043; Invitrogen Corporation, Camarillo, UK). Sections were finally stained with diaminobenzidine (1718096; Roche, Mannheim, Germany), counter-stained with Mayer's hematoxylin, and analyzed using a light microscope (*Micili et al., 2013*).

**Table 1  Comparison of dermatitis scores in study groups.** Values are expressed as the median (25–75 percentile). Two group comparisons were made using Mann Whitney $U$ test.

| Variables | Group I control | Group II vehicle control | Group III resveratrol | P value[a] |
|---|---|---|---|---|
| | Mean ± SD | Mean ± SD | Mean ± SD | |
| | Median(IQR) | Median(IQR) | Median(IQR) | |
| 5th week | 0.57 ± 0.53[b] | 8.29 ± 0.49 | 8.42 ± 0.79 | 0.001 |
| | 1.0 | 8.0 | 8.0 | |
| | (0.0–1.0) | (8.0–9.0) | (8.0–9.0) | |
| 6th week | 0.57 ± 0.53 | 8.86 ± 0.69 | 5.14 ± 1.68[c] | 0.001 |
| | 1.0 | 8.0 | 4.0 | |
| | (0.0–1.0) | (8.0–10.0) | (4.0–7.0) | |

**Notes.**

[a] $P$ value was calculated by Kruskall Wallis $H$ test.

[b] Significantly lower compared to Group II and Group III.

[c] Significantly lower compared to Group II.

IQR, Interquartile Range; SD, Standard deviation.

## Semi-quantification of immunostaining

For each animal two adjacent sections were taken. Five images per section/animal were evaluated and the average immunoscoring of these images were calculated. Each section was graded by two blinded histologists to maintain consistency of the scoring system. A grading system was used to score the quantity of anti-IL-33, anti-IL-25, anti-TSLP, anti-caspase-3 positive staining in the sections. Semi-quantitative score was defined as follows: mild $(+)$, moderate $(++)$, strong $(+++)$ and very strong $(++++)$ brown staining. Staining intensity was graded semiquantitatively using H-scores, which were calculated using the following equation: H-score $= \Sigma \mathrm{Pi}\,(i+1)$, where i was equal to the intensity of immunohistochemical staining with a value of 1–4, and Pi was the percentage of epithelial cells stained with each intensity, varying between 0–100% (*Yuksel et al., 2008*).

## Statistical analysis

Values are presented as the mean ± standard deviation (SD). Normality of the distribution was assessed using the Kolmogorov–Smirnov test. The measurements followed a non-normal distribution, therefore non-parametric comparisons were made by the Kruskal–Wallis test. Pairwise comparisons were made using the Mann–Whitney $U$-test. A $p$ value less than 0.05 was considered significant.

## RESULTS

### Dermatitis score and body weight

All mice developed AD-like lesions with repeated DNFB challenge in both group II and III at the end of 5th week. The application of DNFB to the shaved back of mice firstly induced erythema and hemorrhage, then edema, erosion, excoriation, dryness and scaling appeared. Dermatits scores were not significantly different between groups. Treatment with resveratrol during 6th week resulted a decreased dermatitis score in group III that is significantly lower compared to group II (Table 1 and Fig. 2). There was no significant difference between groups in aspect of body weight gain (data not shown).

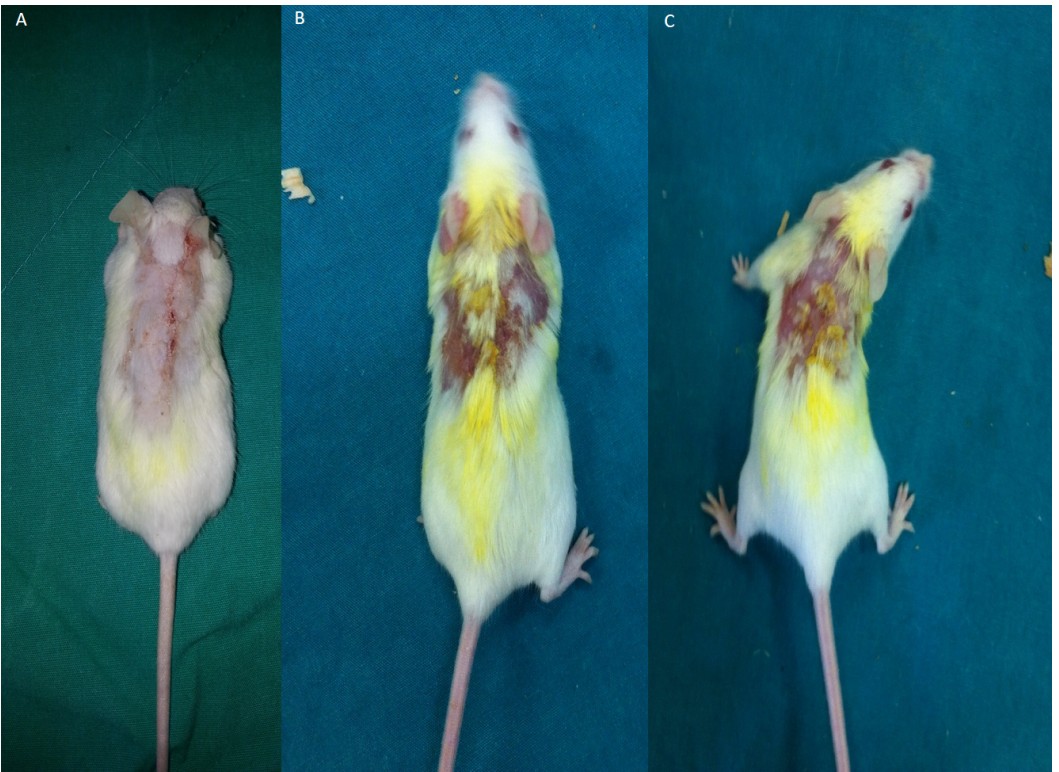

**Figure 2** **Representative pictures of dermatitis in groups after a 1-week treatment.** (A) Control group 1, Erythema/haemorrhage:1/3; 2, Scaling/dryness:0/3; 3, Edema:0/3; 4, Excoriation/erosion:0/3.DS:1. (B) Vehicle control group. 1, Erythema/haemorrhage:1/3; 2, Scaling/dryness:3/3; 3, Edema:1/3; 4, Excoriation/erosion:3/3.DS:8 (C) Resveratrol treatment 1, Erythema/haemorrhage:1/3; 2, Scaling/dryness:1/3; 3, Edema:0/3; 4, Excoriation/erosion:2/3.DS:4.DS; dermatitis score.

## Histological evaluation

Epidermal thickness was significantly greater in group II (97.39 ± 23.26 µm) than in group I (20.28 ± 1.15 µm) ($p < 0.05$), indicating that the model for AD-like skin lesions had been successfully established (Fig. 3 and 4). Epidermal thickness was significantly lower in group III (40.72 ± 12.66 µm) than in group II ($p < 0.05$) (Figs. 3 and 4).

## Immunohistochemical analysis

The number of IL-25 positive cells per field in the skin biopsy were significantly higher in group II than in group I ($p < 0.05$) (Table 2, Figs. 5A, 5B and Fig. 6). The number of IL-33 positive cells in skin tissue were significantly higher in group II than in group I ($p < 0.05$) (Table 2, Figs. 5D, 5E and Fig. 6). The number of TSLP positive cells in skin tissue were significantly higher in group II compared to group I (Table 2, Figs. 5G, 5H and Fig. 6). The number of caspase-3-positive cells, as an indicator of apoptosis, was significantly higher in group II than in group I in skin biopsy ($p < 0.05$) (Table 1, Figs. 5J, 5K and Fig. 6).

The number of IL-25 positive cells were significantly lower in group III compared to group II in skin biopsy (Table 2, Figs. 5B, 5C and Fig. 6). IL-33 positive cells significantly lower in group III compared to group II (Table 2, Figs. 5E, 5F and Fig. 6) in skin tissue.

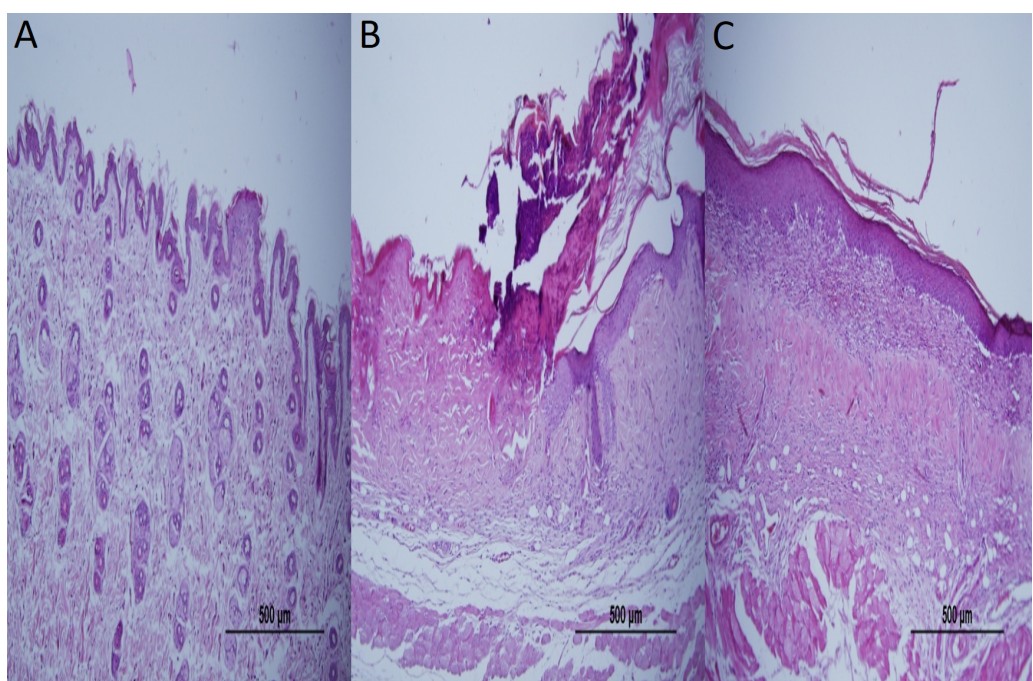

**Figure 3** **Representative H&E staining of skin tissues in groups after 1-week treatment.** (A) Control group; Normal regular epithelium. (B) Vehicle control group; Thickening of the epidermis and epidermal irregularity. (C) Resveratrol treatment; A minimally irregular epithelium accompanying epithelial thickness.

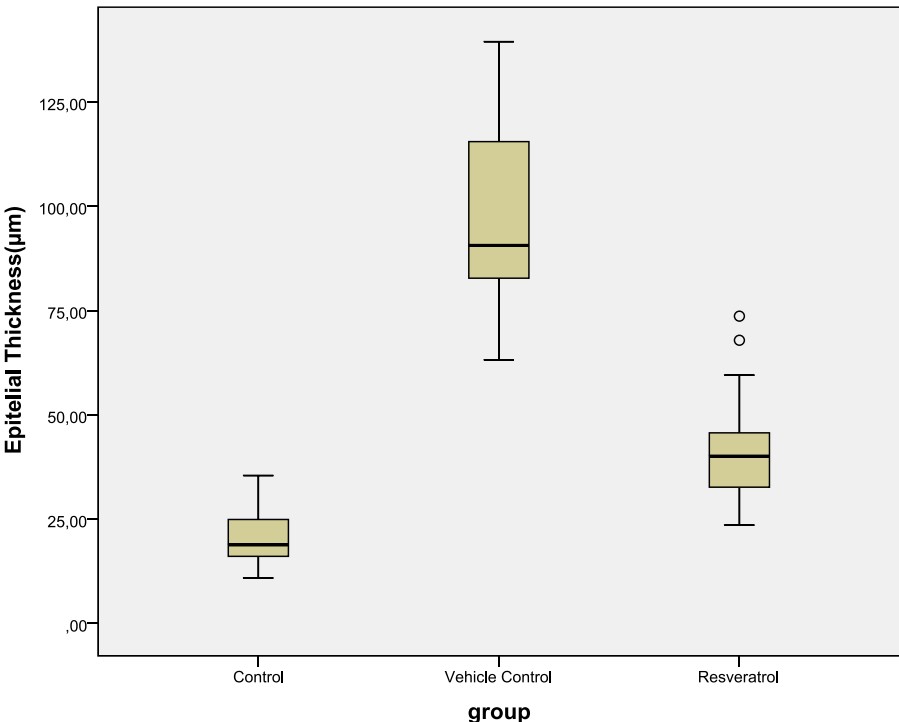

**Figure 4** **Boxplot of the epidermal thickness ($\mu$m) in groups.**

**Table 2 Comparison of *H* scores in study groups.** Values are expressed as the median (25–75 percentile). *P* values were calculated using the Kruskal Wallis and Mann Whitney *U* test.

| Variables | Group I control | Group II vehicle control | Group III resveratrol | *P* value[a] |
|---|---|---|---|---|
| | Mean ± SD Median(IQR) | Mean ± SD Median(IQR) | Mean ± SD Median(IQR) | |
| IL-25 | 164.85 ± 33.37 149 (138.5–206.5) | 297.50 ± 28.83[b] 286 (277.75–309.0) | 234.79 ± 34.98[b,c] 226 (210–246) | 0.001 |
| IL-33 | 167.21 ± 39.48 149 (138.5–206.5) | 311.71 ± 60.13[b] 312.5 (276.50–312.50) | 247.71 ± 36.59[b,c] 230 (213.25–285.75) | 0.001 |
| TSLP | 177.5 ± 32.76 198 (141–206) | 301.93 ± 69.86[b] 297.92 (229.75–367.25) | 247.5 ± 45.30[b,c] 216.5 (207.75–296.25) | 0.001 |
| Caspase-3 | 157.07 ± 36.43 134 (129–202.5) | 282.42 ± 37.41[b] 288.5 (275–292.50) | 214.57 ± 22.01[b,c] 206 (202.75–217.25) | 0.001 |

**Notes.**

[a] *P* value was calculated by Kruskall Wallis *H* test.
[b] Significantly higher compared to Group I.
[c] Significantly lower compared to Group II.
IL, Interleukin; TSLP, Thymic stromal lymphopoietin..

Number of TSLP positive cells in skin biopsy were lower in group III in compared to group II (Table 2, Figs. 5H, 5I and Fig. 6). The number of caspase-3-positive cells in skin biopsy was lower in group III than in group II (Table 1, Figs. 5K, 5L and Fig. 6).

## DISCUSSION

Atopic dermatitis is a relapsing, highly pruritic chronic inflammatory disease of the skin that is associated with significant morbidity and has deleterious effects on the quality of life of patients. It also places a substantial financial burden on both the patient's family and society. The clinical presentation of AD includes erythematous, pruritic, and lichenified skin on some parts of the body (*Lee & Detzel, 2015*). The early onset of AD in infancy often triggers the atopic march, which leads to the sequential development of asthma and allergic rhinitis. It is thus the initial step towards subsequent allergic diseases, therefore making an accurate diagnosis and providing appropriate treatment are critical. The pathophysiology of the disease is complex, as it involves impaired epidermal barrier function, a *T*-cell-mediated inflammatory skin reaction, and accompanying keratinocyte apoptosis (*Werfel, 2009*). The mainstay therapies of AD are topical emollients to provide an effective epidermal barrier, the avoidance of triggers, and anti-inflammatory therapy with topical corticosteroids (TCSs) or topical calcineurin inhibitors (TCIs) (*Weidinger & Novak, 2015*). However, regular long-term use of TCSs can lead to suppression of the hypothalamic-pituitary-adrenal axis, growth retardation in children, glaucoma, and skin atrophy, while the use of TCIs might increase the risk of lymphoma (*Dhar, Seth & Parikh, 2014*; *Hui et al., 2009*). Besides these side effects, in a subgroup of patients with severe AD,

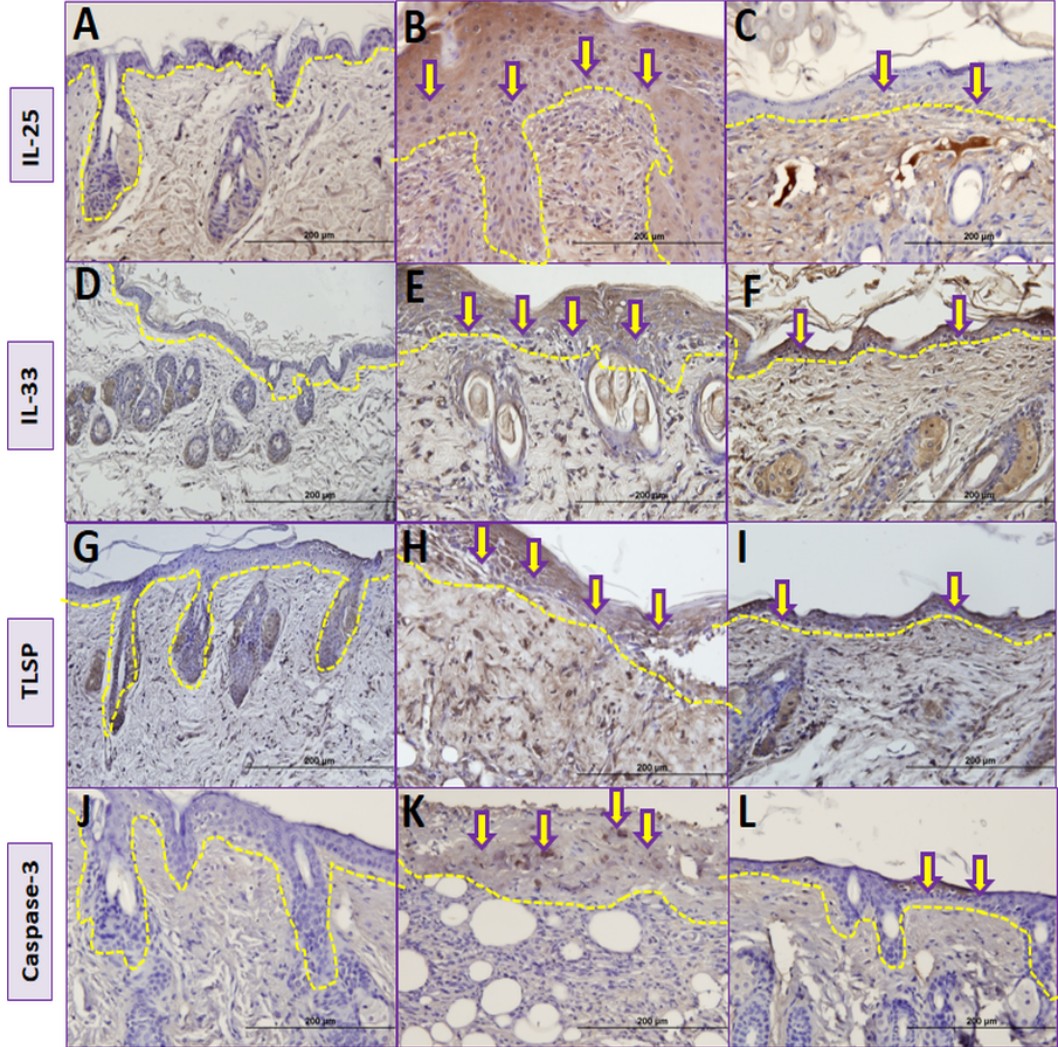

**Figure 5** **Comparison of immunohistochemical analysis between groups.** (A, D, G, J) Control group; (B) Vehicle control group; yellow arrows shows prominent immunostaining for IL-25. (E) Vehicle control group; yellow arrows shows prominent immunostaining for IL-33. (H) Vehicle control group; yellow arrows shows prominent immunostaining for TSLP. (K) Vehicle control group; yellow arrows shows prominent immunostaining for caspase-3. (C) Resveratrol treatment group; yellow arrows shows lower immunostaining for IL-25. (F) Resveratrol treatment group; yellow arrows shows lower immunostaining for IL-33 (I) Resveratrol treatment group; yellow arrows shows lower immunostaining for TSLP (L) Resveratrol treatment group; yellow arrows shows lower immunostaining for caspase-3. The dashed lines indicate the approximate location of the epidermal basement membrane. IL, interleukin; TSLP, thymic stromal lymphopoietin.

these topical therapies are insufficient to control symptoms, therefore systemic treatment options become necessary. Immunosuppressant agents such as systemic corticosteroids, cyclosporine, and azathioprine should be considered in case the disease activity cannot be adequately controlled with conventional topical treatments. These systemic treatments can have serious, even life-threatening adverse effects, mainly due to immunosuppression (*Ricci et al., 2009*). However, there is still no cure for AD, therefore new systemic treatment options with minimal side effects are in demand.

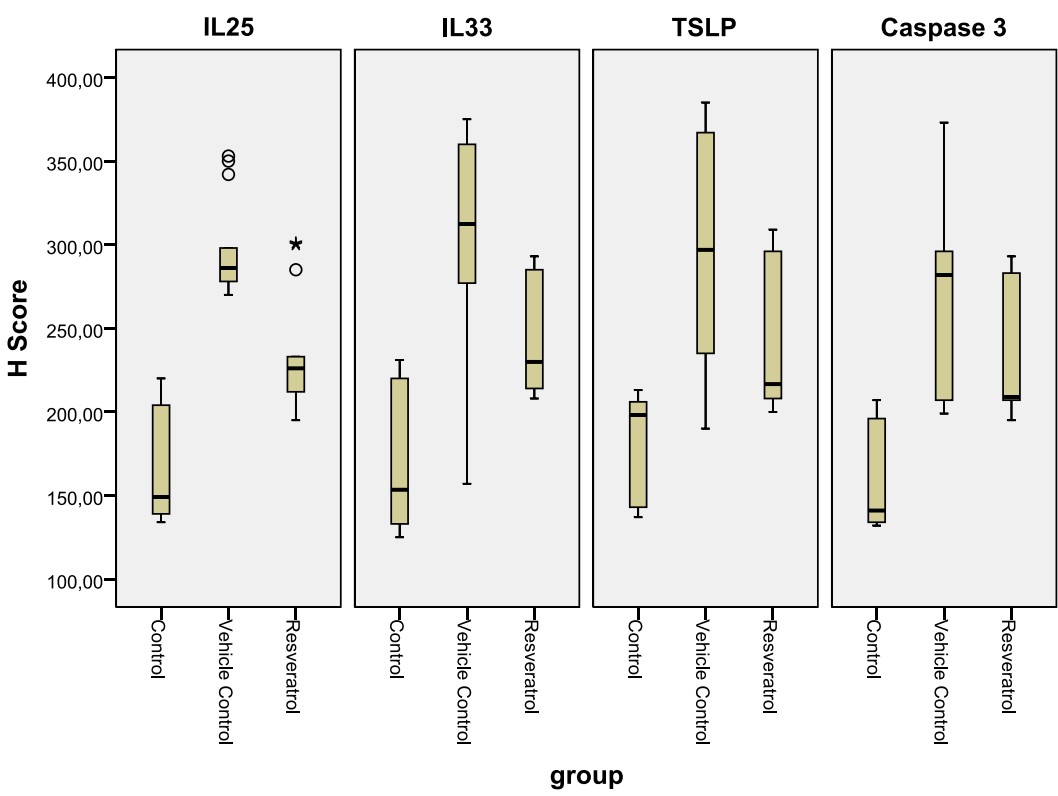

**Figure 6  Boxplot of the IL-25, IL-33 and TSLP H scores in the various groups.** IL, interleukin; TSLP, thymic stromal lymphopoietin.

In this study, we investigated the effects of resveratrol on epidermal thickness, keratinocyte apoptosis, and keratinocyte-derived cytokines on AD-like skin lesions. We examined the thickness of the epithelium and immunohistochemical staining of IL-25, IL-33, and TSLP antibodies to assess the severity of inflammation; we also used immunohistochemical analysis of caspase-3 to assess apoptosis. In this study, we found that resveratrol was effective at ameliorating AD-like lesions by controlling keratinocyte-derived inflammation and keratinocyte apoptosis.

In previous research, the role of resveratrol as a treatment modality for allergic diseases was investigated, and many important biological pathways were identified through animal studies. Resveratrol exerted anti-inflammatory effects on a murine model of eosinophilic chronic rhinosinusitis with nasal polyps by inhibiting lipoxygenase pathway and eosinophil recruitment (*Kim et al., 2013*). Treatment with systemic resveratrol improved chronic structural airway changes such as subepithelial extracellular matrix thickness and fibrosis, with decreased expression of transforming growth factor beta-1 (TGF-$\beta$1) (*Royce et al., 2011*). Resveratrol treatment also caused decreased mast cell degranulation and allergic inflammation by suppressing monocyte chemotactic protein-1 and macrophage inflammatory protein-2 in a mouse model of passive cutaneous anaphylaxis (*Han et al., 2013*). In a murine model of asthma, resveratrol treatment showed beneficial effects on mitochondrial function and attenuated oxidative stress (*Reddy, 2011*). A recent

experimental study investigated the effects of resveratrol on house-dust-mite-induced AD in mice, showing that resveratrol treatment down-regulated high-mobility group box (HMGB)1, which is secreted by various immune cells and acts as an important mediator in chronic inflammatory diseases. HMGB1 binds to its receptor, which in turn activates nuclear factor kappa-light-chain-enhancer of activated B cells (NF$\kappa$B). It was thus suggested that NF$\kappa$B may play a role in the transformation of environmental insults into inflammation in skin tissue (*Karuppagounder et al., 2014*). Although all of these studies have shown that resveratrol has many favorable effects on allergic diseases, to the best of our knowledge, no studies have investigated its effects on keratinocyte-derived cytokines and keratinocyte apoptosis in experimental AD-like lesions.

Atopic dermatitis is elicited by the interplay between various cell types, including *T*-cells, Langerhans cells, basophils, eosinophils, and keratinocytes. Among these cell types, keratinocytes have attracted more attention in the pathogenesis of AD due to their roles in skin barrier function and their contribution to the initiation and maintenance of inflammation (*Holgate, 2007*). Epidermal thickening is obvious in chronic skin lesions of AD, as demonstrated clinically with lichenified plaques and microscopically with acanthosis. *Wu et al. (2014)* found that resveratrol inhibited normal human epidermal keratinocytes by down-regulating aquaporin-3 in a concentration-dependent manner. In our study, resveratrol treatment led to smaller epidermal thickness, provided a regular epithelium, and produced clinical improvements in AD-like skin lesions.

There has been more supporting evidence that keratinocytes act as true innate immune cells. Keratinocytes secrete several inflammatory mediators that exert a variety of local and distant effects (*Esche, De Benedetto & Beck, 2004*). IL-25, IL-33, and TSLP are generated by epithelial cells including keratinocytes as well as other cells, and considerable evidence has suggested that these cytokines play a crucial role in the induction of both innate and adaptive Th2 responses.

The pathologic mechanism behind AD begins with the processing of allergens by local dendritic cells (DCs), which migrate into draining lymph nodes. These DCs initiate the differentiation of prime naive allergen-specific CD4 cells into Th2 lymphocytes, which secrete various cytokines (*Leung et al., 2004*). In this context, the following question arose: how do these DCs become activated to start inflammation in AD? It was shown that TSLP released from keratinocytes could be an activator of DCs. In addition, TSLP has been demonstrated to support the migration, maturation, and activation of DCs in AD skin lesions. Infectious agents and their products, allergens, trauma, and some cytokines, could induce TSLP expression in skin cells. TSLP receptors have been identified on cell types involved in immunological responses, such as *T*-cells, B-cells, monocytes, mast cells, and natural killer cells (*Cianferoni & Spergel, 2014*). The role of TSLP in AD pathogenesis has been investigated in many studies. *Yoo et al. (2005)* found that overexpression of TSLP in the epidermis led to an AD-like disease in mice. It was also reported that single nucleotide polymorphisms of TSLP and its receptors are associated with AD (*Hunninghake et al., 2010*). The role of TSLP in AD was demonstrated when increased TSLP levels were observed in lesional skin of AD patients, but not in either nickel-induced contact allergic dermatitis or in skin changes associated with lupus erythematosus (*Soumelis et al., 2002*). In

our study, the number of cells that stained positively for the TSLP antibody was increased in AD-like lesions, demonstrating that these lesions resemble AD. In another mouse model, deficiency in notch signaling, which is an important regulator of skin epidermal integrity in keratinocytes, resulted in chronic skin changes and caused high levels of TSLP in keratinocytes (*Dumortier et al., 2010*). In addition to the many studies suggesting a critical role for TSLP in the immunopathogenesis of AD, it has attracted substantial attention as a therapeutic target. It was shown that a traditional Korean medicine known as Naju Jjok inhibited the expression of TSLP by blocking the caspase-1 signaling pathway in DNFB-induced AD-like lesions (*Han et al., 2014a*). Another natural anti-inflammatory agent, tryptanthrin, suppressed TSLP in 2, 4-DNFB-induced AD-like skin lesions of NC/Nga mice and inhibited the mRNA expression of TSLP through blockade of the receptor-interacting protein 2/caspase-1/nuclear factor-$\kappa$ B pathway in an activated human mast cell line (*Han et al., 2014b*). Against this background, we hypothesized that resveratrol, which has been shown to be an anti-inflammatory molecule, might affect this key cytokine of AD. In our study, systemic resveratrol treatment was associated with lower expression of TSLP in AD-like skin lesions.

IL-33 is a member of the IL-1 cytokine family. Allergens, microbes, and pro-inflammatory cytokines can trigger the release of IL-33 from the epidermal barrier (*Cevikbas & Steinhoff, 2012*). Its receptor, ST2, presents on various cells including innate lymphoid cells, contributing to the initiation and maintenance of allergic inflammation. It has been shown that ST2 gene polymorphisms are related to the presence of AD and the IL-33-ST2 complex plays a crucial role in AD pathogenesis (*Shimizu et al., 2005*). Transgenic mice with increased skin-specific expression of IL-33 developed AD-like cutaneous manifestations through the activation of innate lymphoid cells in the skin and lymph nodes (*Imai et al., 2013*). In addition, *Savinko et al. (2012)* found increased expression of IL-33 in the epidermis of AD patients. Although these previous studies clearly suggested a pivotal role for IL-33 in the pathogenesis of AD, to our knowledge, this is the first study to investigate the effects of resveratrol on IL-33 expression in AD-like lesions. In our study, resveratrol treatment resulted in lower immunohistochemical expression of IL-33 in the epidermis of AD-like skin lesions compared with that in a placebo group. This finding may provide a treatment option by suppressing one of the initiators of inflammation in AD.

IL-25 is a member of the IL-17 cytokine family that is expressed in epithelial cells in response to proteases such as allergen proteases, trypsin, and papain. It was reported that administration of IL-17 to mice promoted allergic inflammation by inducing IL-4, IL-5, and IL-13 gene expression (*Fort et al., 2001*). Moreover, DCs activated by TSLP enhance allergy-promoting Th2 memory cells by increasing the number of their receptors for IL-25. This could explain the potential role of IL-25 in the regulation of Th2 memory cells (*Wang et al., 2007*). IL-25 was shown to suppress filaggrin expression, resulting in epithelial barrier disruption. Conversely, an impaired epithelial barrier could induce the release of IL-25, which would further worsen epithelial barrier function due to its negative effects on filaggrin (*Hvid et al., 2011*). In terms of the results of this study, treatment with systemic resveratrol led to lower expression of IL-25 in the epithelium of AD-like lesions. Recent studies have indicated a central role for IL-25 in the immunopathogenesis of AD, but, to

the best of our knowledge, this is the first study showing the beneficial effects of resveratrol on IL-25 expression in a mouse model of AD-like skin lesions.

Apoptosis is an essential physiologic process in the establishment and maintenance of both innate and adaptive immunity. However, it also actively participates in inflammatory and immunologic diseases such as asthma and AD (*Trautmann et al., 2000*). Keratinocyte apoptosis was found *in situ* in lesional eczematous skin and patch-test lesions of AD (*Akdis et al., 2001*). It was also reported that interferon-gamma (IFN-$\gamma$)-induced apoptosis in keratinocytes was increased in the skin of patients with AD compared with that in healthy subjects (*Rebane et al., 2012*). $T$-cell-mediated keratinocyte apoptosis via the Fas ligand decreased the expression of the adhesion molecule E-cadherin (*Trautmann et al., 2000*). This resulted in spongiosis, one of the histologic hallmarks of AD (*Trautmann et al., 2001b*). Keratinocyte apoptosis initiates the release of chemotactic factors and promotes $T$-cell infiltration into the epidermis. These $T$-cells increase the key elements of apoptosis, such as interferons and Fas (*Klunker et al., 2003*). The crucial role of keratinocyte apoptosis in inflammation makes it a highly attractive therapeutic target for the treatment of AD. Because of this, we hypothesized that systemic resveratrol treatment might exert anti-inflammatory effects by acting on keratinocytes.

The aspartate-specific cysteine protease (caspase) cascade is considered the main pathway by which apoptosis is orchestrated. The most prevalent protease in the cell is caspase-3. This caspase is the central executioner caspase, which is responsible for the majority of the effects in cellular death (*Zimmermann & Green, 2001*). It was demonstrated that dexamethasone inhibited caspase-3 and caspase-7 and suppressed epithelial apoptosis. Blockage of apoptosis is one of the possible anti-inflammatory effects of steroids (*Trautmann et al., 2001a*). Keratinocytes are vulnerable to caspase-dependent apoptosis in response to IFN-$\gamma$ when the Fas receptor levels increase to a certain threshold (*Tian et al., 2014*). In this study, resveratrol treatment showed beneficial effects on keratinocyte apoptosis, which was demonstrated with lower caspase expression in AD-like lesions. This study supports previous findings showing that apoptosis has an important role in the pathogenesis of AD and indicates its potential importance as a target for treatment.

There are some limitations to this study. First, although we demonstrated beneficial effects of resveratrol on inflammation and apoptosis, we could not reveal the molecular pathways by which resveratrol acts on the epithelium of AD-like lesions. Inhibition of the expression of NF$\kappa$B is a possible common pathway because this transcription factor both activates the cytokines involved in Th2 inflammation and regulates the genes affecting apoptosis (*Barkett & Gilmore, 1999*; *Makarov, 2000*). In addition, the expression of NF$\kappa$B has been found to be increased in the epithelium in chronic inflammatory diseases such as asthma (*Donnelly et al., 2004*). *Ren et al. (2013)* demonstrated the suppressor effects of resveratrol on NF$\kappa$B signaling. Even we could not make a clear connection with $T$ cell response and apoptosis, the inhibitor effects of resveratrol on NF$\kappa$B expression might resulted anti-inflammatory and anti-apoptotic effects in this AD-like murine model. Moreover, *Feng et al. (2002)* showed that low dose resveratrol treatment led to a Th1 dominant immune response with enhanced expression of IL-2, IFN-$\gamma$ and IL-12. However, we could not show the effect of resveratrol on the Th-1 derived cytokines which

should be taken into account in future experimental studies. Second, two blind histologists evaluated epithelial thickness in order to avoid a potential bias in our study but a marker for cell proliferation such as Kİ-67 protein could give a more conclusive data in this aspect (*Scholzen et al., 2002*). Third, this study was conducted on mice and the findings cannot be reliably extrapolated to AD in humans.

## CONCLUSION

In conclusion, our data suggest that systemic resveratrol treatment exerts anti-inflammatory and antiapoptotic effects in a murine model of AD-like lesions. Although it is too early to draw definitive conclusions, our data indicate that resveratrol may be therapeutically beneficial to improve epithelium-derived allergic responses. Specifically, it may be effective at suppressing the very first step in inflammation.

### Funding
The authors received no funding for this work.

### Competing Interests
The authors declare there are no competing interests.

### Author Contributions
- Sule Caglayan Sozmen conceived and designed the experiments, performed the experiments, analyzed the data, wrote the paper, prepared figures and/or tables, reviewed drafts of the paper.
- Meral Karaman and Serap Cilaker Micili conceived and designed the experiments, performed the experiments, analyzed the data, wrote the paper, reviewed drafts of the paper.
- Sakine Isik, Zeynep Arikan Ayyildiz and Alper Bagriyanik analyzed the data, contributed reagents/materials/analysis tools, reviewed drafts of the paper.
- Nevin Uzuner and Ozkan Karaman conceived and designed the experiments, contributed reagents/materials/analysis tools, reviewed drafts of the paper.

### Ethics
The following information was supplied relating to ethical approvals (i.e., approving body and any reference numbers):
   Dokuz Eylul University, Animal Ethics, 92/2013.

### Data Availability
   Figshare: https://figshare.com/s/74717c3f678fc983982f.

### Supplemental Information
Supplemental information for this article can be found online at http://dx.doi.org/10.7717/peerj.1889#supplemental-information.

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
