# Peer review of "Resveratrol ameliorates 2,4-dinitrofluorobenzene-induced atopic dermatitis-like lesions through effects on the epithelium"

_PeerJ, doi:10.7717/peerj.1889_

## Round 0.1 · original submission · Major Revisions

Your manuscript has been reviewed by two subject experts who have raised significant concerns about the study model and quality of images. While the conclusions of your study appears to be substantiated by your data, these major concerns do not enable the manuscript to be accepted in its current form. If you are able to address the reviewer comments explicitly, we would be happy to reconsider your work.

Reviewer 1 ·

Basic reporting

In this study, the authors show that resveratrol can ameliorate the development of lesions in a model of dermatitis. The data presented in figures and table is convincing. Unfortunately, the manuscript requires a lot of revision as it is confusing and the description of the model system is inadequate. As it is written, the text of the manuscript does not do justice to the scientific findings.

Major comments:
1. The description of the model is inadequate and very confusing.
a. The “Induction of dermatitis and experimental schedule” in Methods should be rewritten in greater detail.
b. Figure 1 should be re-drawn to depict all 3 groups of mice.

2. In Results, it is suggested that DMSO itself causes lesions. It seems that this is due to the inappropriate description of the model and of the 3 groups of mice studied. This is a major problem with the manuscript. This must be corrected. Otherwise, the manuscript does not make sense.

3. In Results, the authors should direct the reader to specific Figures rather than to all Figures at once. For example, in Lines 129 and 132 the reader is directed to Figure 2, Figure 3 and Figure 4 all at once. Also, in Lines 134 and 139, the reader is directed to Table 1, Figure 4 and Figure 5. Each experiment and accompanying figure should be thoroughly described within the body of the manuscript.

4. Figure Legends should be more descriptive.

5. Figures should be higher quality.

Experimental design

The experimental design is appropriate. However, as stated in Basic Reporting, the description of the model system and of the animal groups studied is inadequate and confusing; thus, requiring major revision.

Validity of the findings

The reported scientific findings are substantiated by the data provided in the Figures. Thus, the findings are valid.

Additional comments

I think your finding that resveratrol can ameliorate the development of dermatitis-like lesions is valid and substantiated by the well-reported anti-inflammatory properties of the polyphenol. However, the description of the model system and especially of the three groups of mice studied is very confusing. The Results section should be extended to thoroughly describe the strategy and findings from each experiment. In doing so, the reader can be directed to only one Figure or Table at a time. I think that your scientific findings are worthy of publication once these revisions are made.

Reviewer 2 ·

Basic reporting

There is an issue with resolution of the figures.

Experimental design

There is a lack of clarity in the experimental methods. For example, author does not provide evidence as to the gender type of the mice or whether all groups received the carcinogen. I would suspect that one of the groups were untreated, but it wasn't clearly stated. The research does not provide an enhancement of the knowledge gap of the mechanism of action of resveratrol on AD.

Validity of the findings

The author does not provide robust data findings about how resveratrol ameliorates DFNB.

Additional comments

REVIEW:

In the article Resveratrol ameliorates 2,4-dinitrofluorobenzene-induced atopic dermatitis-like lesions through effect on the epithelium, the authors investigated the effects of resveratrol on epidermal thickness, keratinocyte apoptosis, and keratinocyte-derived cytokines on AD-like skin lesions. In addition they examined the thickness of the epithelium and immunohistochemical staining of IL-25, IL-33, and TSLP antibodies to assess the severity of inflammation; also used immunohistochemical analysis of caspase-3 to assess apoptosis. In this study, they found that resveratrol was effective at ameliorating AD-like lesions by controlling keratinocyte-derived inflammation and keratinocyte apoptosis. Although, the authors were able to demonstrate that the humoral response (IL-25. IL-33, TSLP inducing Th2 response) was affected by resveratrol there are some significant flaws to the manuscript. The authors did show an effect of resveratrol on induced apoptosis response through caspase 3, but didn’t make a clear connection with T-cell response by evaluating IL-2 and its connection to AD. In addition, the authors fail to mention that low levels of resveratrol had also been shown to induce an immune response in DNFB mice, demonstrated by Yong-Hong et al. Therefore, the authors could have evaluated the induction of cell-mediated response by resveratrol by examining IFN-gamma, and IL-12. The authors did demonstrate an effect of resveratrol on skin thickness, but fail to provide a correlative to whether the resveratrol had an effect on proliferation which could have been evaluated by Ki-67. As a result of the immune marker gaps, authors failed to provide a clear picture of the mechanism of action by resveratrol on DNFB. Overall, the manuscript lack clarity and did not provide a convincing argument for the mechanism of action. There are other points that need to be clarified as well.


Other points:
1. Authors failed to provide information about incidence of the atopic dermatitis. Did all of the animals develop atopic dermatitis? Also, how many animals per group and were these all male or a mixture of female and male or all female mice?

2. The control group was confusing to me, it wasn’t clear, but I suspect that this group did not receive any DNFB. Please make clear in the method section.

3. What percentage of DMSO was given to the mice?

4. The authors indicated that erythema were significant in group II, but fail to provide a more quantitative measure of the severity of the dermatitis (erythema/hemorrhage, dryness/scaling, edema/swelling, or erosion/excoriation) and it would be affected by resveratrol. This information would give us an idea of what type of lesions were produced, acute lesions or chronic lesions.

5. The authors also failed to assess the body weight and food consumption of the animals which also provides an indirect assessment of the toxicity or lack of toxicity.

6. The authors need to be clear and provide as much uniformity in their tables and figures. For example, figure 3 and table indicate control, vehicle control, and resveratrol and then another indicate vehicle control, placebo, and resveratrol. The problem is that placebo was not explain in the text or in the figure legend of figure 5; is it meant to be equivalent to the control group.

7. There seems to be a slight problem with the resolution in figure 1 and 2.

---

## Round 0.2 · accepted · Accept

The revised version has adequately addressed the concerns raised by the reviewers - Congratulations!

Reviewer 1 ·

Basic reporting

The authors have addressed all of the comments and concerns from the initial review. The data presented in this manuscript is relevant and meaningful as it adds to our knowledge of the anti-inflammatory effects of Resveratrol.

Experimental design

The experimental design sufficiently addresses the question posed by the authors.

Validity of the findings

The findings are valid, and add to our current knowledge of the anti-inflammatory properties of resveratrol. The data supports the author's conclusions.

Additional comments

Thank you for addressing the comments and concerns.